# Nursing Graduates’ Preparedness for Practice: Substantiating the Call for Competency-Evaluated Nursing Education

**DOI:** 10.3390/bs13070553

**Published:** 2023-07-04

**Authors:** Tammy McGarity, Laura Monahan, Katelijne Acker, Wendi Pollock

**Affiliations:** 1College of Nursing, Texas A&M University Corpus Christi, Corpus Christi, TX 78412-5800, USA; tammy.mcgarity@tamucc.edu (T.M.); k4aacker@me.com (K.A.); 2College of Nursing, University of Illinois Chicago, Chicago, IL 60607, USA; 3Department of Social Sciences, Texas A&M University Corpus Christi, Corpus Christi, TX 78412-5800, USA

**Keywords:** nursing practice, nursing education, preparedness

## Abstract

Practice readiness continues to be a challenge in healthcare. This was especially evident during the COVID-19 pandemic. This focused descriptive–correlational study examined nurses’ perceived preparedness for practice during the pandemic. One hundred and eighty-four registered nurses (RN) responded to Qualtrics survey questions addressing the competencies they perceived they had and the competencies they felt they needed that would have better prepared them to care for patients during the COVID-19 pandemic. The results demonstrated that although these nurses felt competent in certain areas, they perceived that they needed more education in those same areas to feel better prepared. Bivariate correlations and linear regression analysis indicated that institutional competency development, education, and work experience influenced perceived competency.

## 1. Introduction

Formal nursing education strives to prepare and graduate nurses who are ready to practice according to employers’ expectations for competent patient care delivery [1,2]. However, readiness for nursing practice is often ill-defined, and this imprecision creates a lack of continuity across the regulatory, academic, and practice sectors, which adds to the incongruencies in expectations for practice readiness [3]. Nursing competency includes the core abilities that are required to understand the needs of the patient, the ability to provide care, the ability to collaborate, and the ability to support decision making; thus, it is important to clearly define nursing competency to establish a foundation for the nursing curriculum [4]. Papathanasious et al. [5] identified critical analysis as a set of questions applied in an event or to a concept for the determination of important information and ideas while discarding the unnecessary ones. Nurses must be prepared with problem solving skills (often interchangeably used with critical thinking skills in the nursing literature) to recognize changes in patient condition, to perform independent nursing interventions, to anticipate orders, and to prioritize. Preparing nursing students who are practice-ready requires an education curriculum focused on core competencies.

AlMekkkawi and Khalil [6] describe practice readiness as having the necessary knowledge and competencies to care for patients safely and independently, and they note that reflective and problem-based learning can enhance students’ critical thinking and problem solving skills. In healthcare, competency is defined as “an observable ability of a health care professional to integrate knowledge, skills, and attitude” [7] (p. 1089). However, the reality is that although most nursing graduates may possess basic nursing knowledge, they need to increase in critical thinking competencies, as well as the commitment, confidence, and professionalism [8,9] required to be practice-ready.

This has been studied, and the results have demonstrated this through previous research that utilized del Bueno’s [10] validated and reliable tool, the Performance-Based Development System (PBDS), which has been used extensively to assess and validate nurses’ potential to meet competency requirements. The PBDS utilizes methods to evaluate critical thinking, interpersonal skills, and technical competencies [10,11]. Del Bueno’s [12] seminal research report in 2001 indicated that only 35% of newly graduated registered nurses were considered safe or ready for practice. Comparatively, aggregated data from a 2017 study that utilized the PBDS indicated that only 23% of newly graduated registered nurses demonstrated entry-level competencies, even after passing the National Council Licensure Exam for Registered Nurses (NCLEX-RN) [13].

Competent nursing practice was critically brought to the forefront during the COVID-19 pandemic [14,15]. This rare global pandemic led to the most intense scrutiny of nursing education and preparedness across multiple settings in recent history [16,17]. Nurses’ experiences, knowledge, skills, and competency to handle this novel infectious disease have been important topics of discussion in the relevant literature [11,18]. In a qualitative study, Badowski and colleagues [19] stated that nurses working in the frontline during the pandemic recommended that key elements, such as teamwork, communication, flexibility, leadership skills, advocacy, and the development of critical thinking, were crucial concepts to be taught and developed in nursing education. One comment made by a nurse in this study was significant and revealing: “I don’t think the technical skills are what’s needed, but the ability to think through the processes” [19] (p. 670), emphasizing the importance of the development of critical thinking skills. Studying senior nursing students’ experiences during the pandemic, Canet-Vélez et al. [20] noted that students requested more education on infection control and personal safety measures, indicating a perceived lack of competency in these basic nursing concepts. This sentiment was reflected in the fact that recent PBDS data demonstrated a continued loss in preparedness among new nurses, and that only 9% of the 2020 graduated nurses were in the competency range for novice nurses [21]. However, education and training after graduation, in addition to increased work experience, further development of critical thinking skills, and an adherence to professionalism, impacted nurses’ development of competency significantly [22]. During the COVID-19 pandemic, nurses found themselves caring for patients with a novel infection; many workarounds were implemented, and even seasoned nurses found themselves wondering how to deliver safe patient care. In this article, the authors will describe the competency gap nurses experience when transitioning from nursing school to nursing practice, as identified during the COVID-19 pandemic, and offer an evidence-based solution for nursing education to address this gap.

## 2. Materials and Methods

Nursing competencies include essential abilities that enable nurses to assess various sources of information and data, and to utilize these in nursing practice to provide competent, comprehensive care to meet patients’ diverse and complex needs effectively and efficiently [23]. Nursing practice proficiency is expressed in the achievement of competencies that go beyond the combination of knowledge and skills; these competencies improve the quality of nursing care, ensure the safety of patients, and reduce the incidence of near misses in the provision of nursing care [24,25]. Imperative to the development of competencies is that nurses are taught critical thinking skills that include analysis, interpretation, inference, and evaluation to make continual enhancements that address ongoing or newly developing patient needs [26]. Professional confidence in critical thinking abilities affects the application of nursing competencies, which can, in turn, impact the quality of patient care given [27].

Nurses’ perception of the competencies they had and those they lacked were the subjects of this research. This study aimed to investigate nurses’ perception of their competency to practice during the novel infectious disease pandemic of COVID-19. The following research questions were the basis of this study: which competencies did nurses perceive to have that prepared them for practice during the COVID pandemic, and which competencies did nurses perceive they lacked that would have better prepared them for practice? Additionally, did nurses perceive that their education and/or experience prepared them for practice during the COVID-19 pandemic?

This focused descriptive–correlational study addressed the perceived competency and preparedness of nurses employed during the height of the COVID-19 pandemic during the period of August through October 2020. Invitations to recruit registered nurses to participate were posted on multiple social media sites, including Facebook, Instagram, and LinkedIn. All participants gave informed consent prior to completing the questionnaire. This informed consent was accessed from a link in the invitation and completed prior to completing the Qualtrics questionnaire. An Internal Review Board at a prominent Texas university approved this study.

All registered nurses (RNs) registered in at least one state in the United States were eligible to participate. The invitation was posted on social media sites, and the RNs who responded were from various states across the United States. The sample consisted of one hundred and eighty-four (184) RNs who responded to the invitation, gave informed consent, and completed the survey. Demographic data collected included length of work experience, level of education, current work status, current position held, specialty certification, and institutional type of work setting (Table 1).

The analysis revealed that participants averaged 3.03 years of practice. Most of the respondents (42%) held a Bachelor of Science in Nursing degree, followed by a Master of Science in Nursing degree (35.2%), and lastly, 19% of respondents had an associate degree or diploma in Nursing. The most common positions at work listed were frontline nurses or direct caregivers (55.4%), and 77.7% of the respondents worked full-time at a single facility. Many of the respondents (44.6%) held a specialty certification. Participants practiced in a variety of settings, ranging from teaching hospitals/systems (34.8%) to acute care hospital/medical centers (35.9%) and outpatient clinics (12%).

Because an instrument to measure nurses’ perceived competency was not available, the authors developed an original set of survey questions to investigate how nurses assessed their level of competency in multiple areas during the novel infectious disease pandemic. The survey consisted of multiple choice questions with Likert-type responses (even-point, positive to negative response categories for survey questions) as well as fill-in-the-blank questions, which asked participants for their level of agreement with the statement that their education and competency validations prepared them to care for COVID-19 patients. Participants were asked to list the top five competencies that they felt had prepared them to care for patients during the pandemic, and the top five competencies they perceived that they lacked. The survey also addressed their perceptions of competency preparation, competency validation modes during orientation, and ongoing competency development.

Frequencies were calculated to describe the distributions of the number one competency that nurses perceived had prepared them to care for patients during the pandemic, and the number one competency that nurses perceived that they were lacking that would have better prepared them for practice. Bivariate correlations were completed to examine whether statistically significant relationships existed between the competencies that participants perceived had prepared them, those that would have prepared them better, and the method and timing of the competency validation after graduation. Finally, linear regression was used to examine the impact of multiple variables on respondents’ perception that the competency validation/education they received had prepared them to nurse patients during the COVID-19 pandemic.

How to accurately measure and analyze perceptions and feelings is an ongoing debate in academia because these concepts cannot be measured in a truly continuous (interval or ratio) manner. However, numerous scholars argue that Likert-style measures of concepts are an acceptable way to utilize measurements of perceptions or feelings in higher-level, multivariate, predictive analysis, and that they can be treated as continuous variables with the assumption that there are equal distances between the categories [28,29]. The variable must have at least five answer response categories to be used in this manner [30]. For the dependent variable in the current analysis, nurses were asked to give their level of agreement with the following statement: “The competency validation/education you received prepared you for nursing during the COVID-19 pandemic.” Their responses ranged from strongly disagree (1) to strongly agree (6), with a total of six response categories. Accordingly, this variable is being treated as continuous in the linear regression model. Previous studies have shown that perceptions and feelings demonstrate a link between actions, and that the feelings-to-action relationship appears replicable and linear; additionally, the use of Likert-style survey responses in the measurement of perceptions and feelings is common in social-science-related fields [31].

While linear regression was chosen for presentation in this manuscript, it should be noted that additional, unpresented models were run. In one such model, the dependent variable, nurses’ perception of competency, was recoded into a dichotomous variable (1 = any level of agreement that the respondent felt prepared, and 2 = any level of disagreement that the respondent felt prepared), which was used in a logistic regression model that included all the independent variables in the current paper. Logistic regression is appropriate for a dichotomous dependent variable [32]. All variables that were statistically significant in the linear regression model were also statistically significant in the logistic regression model, and the measures of substantive significance were similar. The lack of divergence in results between these techniques suggests that either is appropriate to use, and linear regression was selected for presentation because it allows for greater detail in the dependent variable.

The independent variables’ nominal data describing participants’ level of experience, education, work status, position, specialty certification, and institutional work setting (see Table 1) were dummy coded or dichotomized. This allowed the variables to be used in linear regression without violating any assumptions of the technique [32].

## 3. Results

Overall, more than 50% of participating nurses perceived that their education did not prepare them for nursing during the COVID-19 pandemic. Only 8% of participants strongly agreed, compared to 10.9% who strongly disagreed, that the education and competency validation they had received had prepared them for nursing during the COVID-19 pandemic.

The results revealed that nurses who participated in in-services and attended competency training classes were statistically significantly more likely to respond that emergency management training had better prepared them to care for patients during the pandemic. Similarly, nurses who had competency training delivered through learning management systems with in-service testing, and through direct observation experiences in COVID-19 patient management, were more likely to respond that they were prepared to care for patients during the pandemic. Additionally, experience at a facility that developed core competencies for high-risk skills, the attainment of a higher level of education, and achievement of more extensive work experience were better predictors for being prepared to care for COVID-19 patients.

Ninety-two (92) individuals responded to the prompt to write in the top five competencies they believed would have better prepared them to care for patients during the pandemic. In contrast, eighty-nine (89) listed the top five competencies covered in their training that they considered to have prepared them. The ‘infection prevention’ topic, which included the use of personal protective equipment (PPE), the donning and doffing skills of PPE, and use of the N95 mask, were the top abilities selected in both categories. Likewise, the ranking of the chosen competencies was similar, with more respondents naming infection control as the top competency that they perceived would have better prepared them (Table 2).

Bivariate correlations were completed to examine whether statistically significant relationships existed between the competencies that participants felt had prepared them and those that would have helped prepare them better and the method and timing of competency validation after graduation. The strongest relationship (*p* = 0.005) indicated that nurses who received no institutional competency development in high-risk skills were more likely to suggest that none of their competencies prepared them to care for COVID-19 patients. Linear regression was used to analyze the effect(s) of the independent variables on nurses’ perceptions of their competency to care for patients during the COVID-19 pandemic. Table 3 illustrates the multivariate model’s statistical (*p* = 0.004) and substantive (R^2^ = 0.363) significance. This suggests that education, workplace, positions, and other elements combined do in fact influence RNs’ perception of competency.

## 4. Discussion

The onslaught of a novel infectious disease has once again drawn attention to the educational preparation of nurses and has led many to question the current landscape in the preparation of nurses to practice [33,34,35]. This study found a perceived lack of academic preparedness in specific, identified concepts. Participants indicated the following elements that influenced their perceptions of competency: working at a facility that developed core competencies for high-risk skills, receiving more education, and having more years of experience. Those nurses in this study who had less experience, worked at facilities with no intense competency orientations, and who cared for COVID-19 patients with only the competencies that they graduated with, stated that they did not feel well-prepared for practice.

COVID-19-related concepts, such as infection control, gas exchange, perfusion, and their related skills, are addressed in every nursing program, yet participants did not feel competent taking care of patients with this novel disease. An example of this is in infection control: although nurses stated that this was the number one competency chosen in multiple categories, the responding nurses in this study indicated a lack of competency in applying infection control standards in the COVID-19 clinical scenarios and applications.

All participants had learned infection control principles during their formal education. Traditional nursing programs teach extensive content, some of which nursing students tend to learn in disease or condition silos. However, basic concepts, such as infection control, gas exchange, perfusion, and their related skills, have been discussed in every nursing school curriculum. However, nursing students and new graduates alike have trouble applying conceptual knowledge, such as those related to infection control, gas exchange, perfusion, etc., across varying disease populations. All or some of these principles could be applied to all infectious patient populations depending on the type of infection or patient, but the basic principles themselves do not intrinsically change. Yet, many participants indicated the need for more instruction in caring for infectious patients, noting that they were unable to apply infection control principles to different patient populations. For example, many reported that they felt competent to perform cardiopulmonary resuscitation (CPR), but that they did not feel competent to apply this skill to a contagious patient. They further expressed that they were comfortable performing basic isolation procedures but that they could not conceptualize and apply the correct applications of PPE in an unknown infectious disease situation.

Analysis of the data indicated that institutional (after graduation) on-the-job instructions and training helped participants feel better prepared. Researchers contend that just-in-time or on-demand education could be an answer to the apparent lack of competencies [36]. However, basic nursing education that nursing students receive should prepare these future RNs to be competent to practice effectively after graduation. This means that these graduate nurses should possess the knowledge and skills to apply them across disease populations. They should effectively display these abilities and apply those critical nursing skills necessary to do the work of nursing.

These findings reinforce Kavanagh and Sharpnack’s [21] assertion that there is an ongoing crisis in nursing competency, and that a vast chasm exists between academia and clinical practice. Crucial factors impacting nursing education today include applicability and accountability for success and what clinical preparedness really requires, which includes keeping current with the rapid pace necessitated in the current learning environments, the need for ongoing education transformations to adequately address contemporary healthcare needs, meeting needs in the face of the rise of rapid technologies and innovations, understanding the impact of digital processing in clinical settings, and addressing the ever-widening gap of inequality in education [21]. Many contend that “warp speed innovation” in healthcare [37] (p. 3) and education’s delayed response to evolving practice innovations continue to cause difficulties [38,39], thereby causing serious detriments in the practice of nursing. Practice readiness has been declining for decades [12,21], and nursing education has struggled to close the practice gap.

Traditional programs moving toward concept-based programs attempt to close this widening gap between academia and practice. Academic and practice partnerships improve teaching and learning applications in a concept-based curriculum that supports competency development and critical thinking skills [40,41]. For example, competency-based education strives to promote innovative approaches to enhance nursing education by utilizing integrated technologies that incorporate active learning and interactive teaching designs to improve efficiency and enhance student learning experiences [21]. Although a concept-based curriculum presents numerous challenges to educational entities in terms of time and faculty support, it effectively promotes student-centered teaching, critical thinking, enriched interactive learning, and the streamlining of essential course content [42].

If competency is expected, then nursing students’ competencies should be continuously and analytically evaluated. The American Association of Colleges of Nursing Baccalaureate Essentials for Nursing Education outlines the expected outcomes of nursing programs [43,44]. The 2021 nursing essentials are formulated as competencies rather than as individual knowledge, skills, and actions (KSAs) [23]. Thus, the American Association of Colleges of Nursing Baccalaureate Essentials incorporate concepts for professional nursing development that are integrated throughout the 2021 nursing essentials, which include such competencies as clinical judgment, communication, diversity, equity, inclusion, ethics, compassionate care, health policy, evidence-based practice, and the social determinants of health [43]. This move requires nursing programs to implement strategies that consistently evaluate graduating nursing students’ competencies and, additionally, should consequently enhance curriculum changes that promote the ongoing development of these necessary critical thinking abilities [11,13,45,46].

The results of this study support the American Association of Colleges of Nursing’s move to nursing essentials that embody the core competencies for nursing education that enhance critical thinking abilities. Likewise, the National Council of State Boards of Nursing demonstrates that critical thinking, problem solving, and critical decision making are interrelated core skills involved in crucial nursing functions that are essential in today’s more complex healthcare environments. The new National Council Licensure Examination for Registered Nurses (NCLEX-RN) format focuses on core competencies related to clinical decision making (by utilizing new types of question formats, such as case studies, that engage students in the decision making process), assessment of developing health assessment cues, prioritization of hypotheses, generation of solutions, and evaluation of outcomes [47]. The development of these core competencies includes the necessary nursing abilities to understand complex patient needs, provide competent and compassionate care, collaborate with other healthcare providers, and support and facilitate decision making [23]. These nursing competencies facilitate the necessary knowledge, skills, and attitudes required to fulfill one’s role as a nurse, regardless of the type of patient.

In the future, if faced with another novel disease, graduates educated under the new American Association of Colleges of Nursing’s essentials should be able to apply required knowledge across various patient populations, demonstrate critical thinking, and exhibit enhanced clinical judgment abilities. These newly graduated nurses should be better prepared for professional nursing practice. Although nursing is a profession that requires lifelong learning, as well as ongoing formation and development, graduates whose competencies are evaluated early on and on a continuous basis should not need to catch up with just-in-time education [36]. They should be ready to acquire new competencies as healthcare continues to change. Various researchers propose that concept-based learning encourages students to synthesize their knowledge more adeptly and globally [42]. However, conceptual understanding alone is not enough. Preparing competent graduates of nursing schools means that they not only have the knowledge to apply resources across disease populations and types of patients, but they also display the behaviors required to do the work of nursing. This study supports the move to competency-based evaluation and the emphasis on continually developing critical thinking skills.

Because an instrument to measure perception was not readily available, the authors, who are all educators and nursing competency experts, developed the questionnaire used for this investigation. A copy of the questionnaire is available in Figure 1. Further utilization of this tool is necessary to demonstrate the validity and reliability of the questionnaire, and its ongoing use is encouraged. The sample size, although small at 184 participants, did represent a variety of workplaces and educational preparations, and would be considered acceptable in that the data collection was completed during the height of the COVID-19 pandemic. Similar investigations are recommended and encouraged to allow for the generalization and applicability of the results.

## 5. Conclusions

This study investigated 184 nurses’ perceptions of their competency during the COVID-19 pandemic and sought to determine whether these nurses perceived that their education and/or experience had prepared them for the public health emergency during the COVID-19 pandemic. More than 50% of the participating registered nurses in this study perceived that their education did not prepare them for nursing during this time, thus demonstrating a perceived lack of academic preparedness in core nursing competencies. This lack of competency preparedness and critical thinking left participants without the skills to translate the application of learned skills to care for this novel disease population. These outcomes support previous studies that utilized the Performance-Based Development System to assess nurses’ potential to meet competency requirements, and which repeatedly demonstrated a perceived lack of competency in basic nursing concepts.

It is imperative to note that practice environments, as well as various regulatory bodies, expect nursing education to prepare nurses to competently practice after graduation. For example, the American Association of Colleges of Nursing emphasizes that the four million nurses in the United States are crucial to the maintenance of patient safety and health-sustaining services. As such, it is critical that the preparation of nurses with consistent sets of core competencies intrinsically enhances the essential partnership between employers and practice leaders in the expectations and provisions of nurses’ obligations of care to their patients [43]. Preparing nurses with a consistent set of identifiable competencies helps employers and the public understand what to expect from nurses, as well as how to distinguish the nursing capabilities in baccalaureate and graduate nursing preparation and their collaborative abilities with other providers of care. Throughout the development of the new 2021 nursing essentials, the American Association of Colleges of Nursing was very intentional in involving both practice leaders and regulatory bodies in the work of infusing innovations and practice expectations into the development of future nursing curriculums and education. The American Association of Colleges of Nursing engaged a variety of practice leaders in this effort to ensure that nursing education remains synchronized with the current and future needs of the healthcare systems, and that it will meet their requisites with the innovations and technologies necessary to meet patient needs effectively [43]. This means that graduate nurses should possess the critical thinking competencies necessary to be skillful and flexible in applying that knowledge across various disease populations. The expectation is that registered nurses should practice those crucial nursing skills that are necessary to perform the work of nursing efficiently and effectively.

Looking further into the nursing education continuum, curriculum changes that encompass competency preparedness and evaluation can effectively improve nursing workforce readiness to care for any type of patient, including those with unexpected or unknown conditions. Nursing orientations, annual evaluations, competency testing, and continuing education should encompass competency evaluations and the assessment of critical thinking skills, as well as the dissemination of institutional knowledge. Closing the education–practice competency gap will require further research that addresses the effect that the ongoing changing landscape of nursing education has on newly graduated nurses’ competencies.

## Figures and Tables

**Figure 1 behavsci-13-00553-f001:**
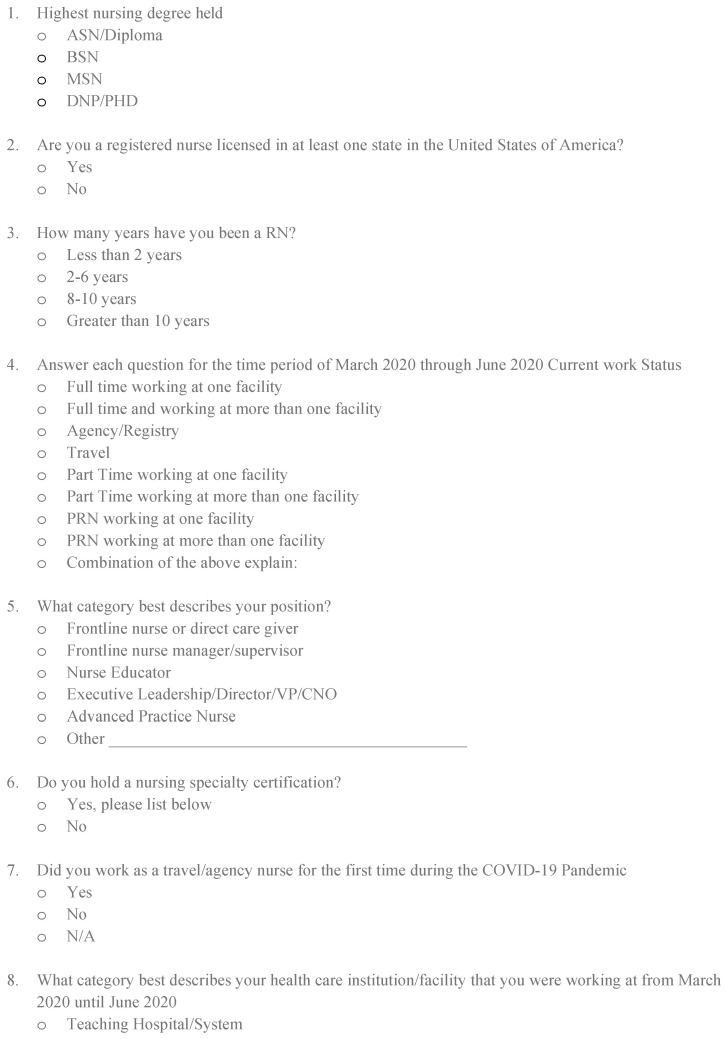
Questionnaire.

**Table 1 behavsci-13-00553-t001:** Overall frequencies of participants.

**Number of Years as an RN**	
Std. Deviation	Min.–Max
1.138	1–4
**Highest Nursing Degree Held**	
Valid Percent	Cumulative Percent
19.2	19.2
41.8	61
35.2	96.2
3.8	100
**Current Work Status**	
Valid Percent	Cumulative Percent
77.7	77.7
7.6	85.3
14.7	100
**Current Position**	
Valid Percent	Cumulative Percent
55.4	55.4
6.5	62
8.2	70.1
6.5	76.6
13.6	90.2
9.8	100
**Specialty Certification**	
Valid Percent	Cumulative Percent
44.6	44.6
55.4	100
**Institution Type** (Where employed from March to June 2020)	
Valid Percent	Cumulative Percent
34.8	37.8
35.9	70.7
12	82.7
17.3	100

**Table 2 behavsci-13-00553-t002:** Competency preparedness perception.

Respondents Felt:	Prepared			Would Have Prepared		
Competency	Frequency	Valid %	Cumulative %	Frequency	Valid %	Cumulative %
Infection Control	27	30.3	30.3	35	38	38
Emergency Management	20	22.5	52.8	17	18.5	56.5
COVID-19 Patient Management	13	14.6	67.4	11	12	68.5
None	10	11.2	78.7	21	22.8	91.3
Other	19	21.3	100	8	8.7	100

**Table 3 behavsci-13-00553-t003:** Linear regression model.

Dependent Variable	Predictive Efficiency	Independent Variables	Unstandardized Coefficients	Standardized Coefficients	Statistical Significance
Agreement from nurses that the competency validation or education they received prepared them for nursing during the COVID-19 pandemic (1 = strong disagreement to 6 = strong agreement that they felt prepared)	F = 2.438 (*p* = 0.004) R^2^ = 0.363 Adjusted R^2^ = 0.214	Highest Degree	0.548	0.298	0.020
		Years as an RN	−0.187	−0.142	0.320
		Work Status-FT in 2+ facilities	0.375	0.067	0.510
		Other Work Status	1.175	0.256	0.011
		Nurse Manager	0.595	0.119	0.260
		Nurse Educator	−0.337	−0.060	0.591
		Executive Leadership	−1.057	−0.211	0.071
		Advance Practice Nurse	0.180	0.046	0.722
		Other Position	1.170	0.194	0.081
		Specialty Cert.	−0.192	−0.065	0.545
		Works at Teaching Hosp.	−0.228	−0.075	0.509
		Works at Clinic	−0.334	−0.078	0.501
		Other Work Location	0.347	0.081	0.468
		Years at Facility	0.127	0.104	0.457
		Competency Prep. On Hire	−0.118	−0.033	0.774
		Competency Prep. Some Other Time	−0.230	-0.057	0.584
		Face to Face competency validation	0.062	0.017	0.869
		Core Competencies for High-Risk Skills	1.177	0.399	0.000

## Data Availability

Not applicable.

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
