# Peer review of "Nursing Graduates’ Preparedness for Practice: Substantiating the Call for Competency-Evaluated Nursing Education"

_behavsci, 2023, doi:10.3390/bs13070553_

Round 1

Reviewer 1 Report

1. The subject is not clear. Does this article describe the gap between academic and practice? Or developing a scale?

2. If this article tried to develop a scale, there has no literatures regarding to develop a scale in introduction section .

3. In Materials and Methods section, there is no literature support for the content of the scale mentioned in the article, no describing about the development process of the scale, and no test reliability and validity.

4. When the reliability and validity of the scale are not clear, the reliability of the statistical results based on this data is questionable.

5. The author tries to introduce the importance of critical thinking ability from the differences between academic and practice, which is a very good thing. However, when the foundation of the previous scale is not reliable, the credibility of the discussed is also questionable.

6. The conclusion should be an integration of the full text, rather than a direct re-description of the above content (such as the characteristics of the participates).

Author Response

Please see attached response to the review.  We appreciate your time and efforts and hope that we have addressed your concerns.  

Reviewer 2 Report

This study deals with important contents that should be reflected in nursing education, so I read the study with interest.

However, the reliability problem of the paper results is presented. This is because the reliability of the measurement tool is not presented, and the reliability of the tool produced by the author is not confirmed. It seems that a lot of supplementation is needed as a thesis that is scientific writing.

The details are presented below.

1. In the introduction part, the presentation of the research background is too brief, and it is necessary to present the necessity and purpose of the research in a clear connection.

2. need a question mark next to Specialty Certification in the table1?

3.Please clearly indicate COVID-19 in Table 2.

4. There's no title for each table.

5. The reliability of the tool is not presented as a whole. 

6. Is there no adjusted R2 value in the regression analysis result?

Author Response

Thank you for reviewing our article.  We hope we have addressed your concerns in the attached document.  

Round 2

Reviewer 1 Report

1. The authors mentioned that they did no develop a scale. Please explain “Since an instrument to measure nurses’ perceived competency was not available, the authors developed an 18-item questionnaire to investigate how nurses assessed their competencies to practice during the novel infectious disease pandemic.” (line 132-134)

2. The author's reply did not answer the reviewer's question. Please confirm the meaning of Linkert's scale.

3. Even when an existing scale was used, the reliability and validity of the scale should be presented in the manuscript. However, the reliability and validity of the scales used was still not presented in the revised manuscript.

4. The problem of reliability and validity of scales have not been resolved, so the credibility of the discussed is also questionable.
